# SNAP: Generalizable Zero-Shot Prediction of Neural Architecture Performance via Semantic Embedding and Graph Learning

## Abstract

Neural Architecture Search (NAS) is a powerful approach to discovering high-performing CNN architectures, but most existing methods incur significant computational costs due to extensive training or sampling. Zero-shot NAS predictors offer an efficient alternative by predicting architecture performance without additional training. However, current methods often yield suboptimal predictions—frequently outperformed by basic metrics like parameter counts or FLOPs—and struggle to generalize across different search spaces or unseen operators. To address these limitations, we propose SNAP(Semantic Neural Architecture Predictor), a novel zero-shot neural predictor that leverages a transformer-based semantic embedding of operator descriptions combined with a Graph Convolutional Network (GCN) for architecture performance prediction. Unlike traditional model-based predictors, SNAP requires only a single initial training phase on NASBench-101, after which it effectively generalizes to arbitrary new search spaces and previously unseen operators without fine-tuning. Extensive experiments across diverse NAS benchmarks demonstrate SNAP's state-of-the-art rank correlation and superior generalization capabilities. Furthermore, SNAP achieves more than $35\times$ search efficiency improvements, discovering competitive architectures with 93.75% CIFAR-10 accuracy on NAS-Bench-201 and 74.9% ImageNet top-1 accuracy on the DARTS space, positioning it as a robust and generalizable foundation for efficient neural architecture search.

## 1 Introduction

Deep convolutional neural networks (CNNs) have achieved remarkable performance across diverse domains, including computer vision, speech recognition, and object detection Krizhevsky et al. (2012); Lin et al. (2015); Qiao et al. (2022); Kaeley et al. (2023). CNN architectures were manually designed, demanding significant time, expertise, and computational resources, particularly when targeting specialized hardware with stringent constraints He et al. (2016); Sandler et al. (2019). Neural Architecture Search (NAS) systematically explores optimal architectures given performance and computational constraints Lin et al. (2020); Zoph & Le (2017). Early NAS methods relied predominantly on reinforcement learning (RL) or evolutionary algorithms, requiring extensive sampling, training, and evaluation, thus incurring substantial computational costs Liberis et al. (2021).

To reduce these costs, zero-cost proxies have been proposed, providing lightweight performance estimates through fast computations like forward passes on small minibatches, gradient-based indicators, or kernel methods White et al. (2023); Krishnakumar et al. (2022); Mellor et al. (2021); Qiao et al. (2024a;b). Despite their efficiency, zero-cost proxies frequently underperform simple heuristics like parameter count or FLOPs, suffer from data dependency, and generalize poorly across diverse search spaces. Additionally, their reliance on similar information sources results in correlated and redundant predictions.

Model-based predictors have emerged as an alternative, employing trained machine learning models—such as Gaussian processes Lévesque et al. (2017) or deep neural networks Shi et al. (2020)—to forecast architecture performance based on structural attributes. Although more accurate, these methods typically require extensive training data from numerous fully evaluated architectures, mak-

ing them computationally costly and limiting their generalization to new search spaces or unseen operators without extensive retraining.

To bridge this gap, we introduce SNAP (Semantic Neural Architecture Predictor), a novel zero-shot neural predictor explicitly designed to generalize robustly to unseen operators and search spaces without retraining. SNAP uniquely combines transformer-based semantic embeddings of textual operator descriptions with a Graph Convolutional Network (GCN) predictor, effectively merging the advantages of zero-cost proxies and traditional model-based predictors. To facilitate future research, we provide open-source access to our implementation.

The main contributions of our work are:

- We propose **SNAP**, a universally applicable zero-shot neural predictor capable of generalizing effectively to unseen operators and new search spaces without retraining. SNAP integrates transformer-based semantic embeddings, DAG-based architectural representations, and a GCN predictor.

- SNAP is comprehensively evaluated across 12 NAS benchmarks, demonstrating superior rank correlation, remarkable generalizability, and independence from traditional proxies.

- Extensive experiments confirm SNAP's state-of-the-art predictive performance and efficiency, achieving more than $35\times$ speedup over existing zero-shot methods and traditional NAS approaches. SNAP identifies competitive architectures achieving 93.75% accuracy on CIFAR-10 (NAS-Bench-201) and 74.9% top-1 accuracy on ImageNet (DARTS).

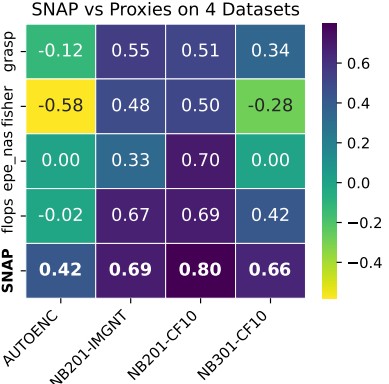

Figure 1: Spearman's correlation between SNAP and 4 other proxies (the higher the better).

## 2 BACKGROUND AND RELATED WORK

### 2.1 NEURAL ARCHITECTURE SEARCH (NAS)

NAS is an important technique in automating the design of neural architectures for a given task White et al. (2023). A typical NAS contains a **search strategy** that selects candidate architectures from a predefined **search space** and an **estimation strategy** that enables the performance estimation for candidates.

**Search space** can be categorized as the following types: macro search space Kandasamy et al. (2019), chain-structures search space Sandler et al. (2019), cell-based search space Liu et al. (2018c); Ying et al. (2019); Dong & Yang (2020), and hierarchical search space Liu et al. (2018b). Among those, cell-based search space is the most popular one in NAS White et al. (2023). The searchable cells (a directed acyclic graph (DAG) of operations) make up the microstructure of the search space while the macrostructure (that defines the number of cells and how they stack together) is fixed. For example, NAS-Bench-101 Ying et al. (2019) contains 423,624 unique architectures, and each cell consists of 7 nodes (each node is chosen from three operators). In NAS-Bench-201 Dong & Yang (2020), there are 15,625 cell candidates and each cell consists of four nodes (each node chosen from five operators). In contrast, the DARTS search space Liu et al. (2018c) is more expansive, featuring approximately $10^{18}$ architectures. It consists of two cells, each containing seven nodes. The first two nodes receive inputs from previous layers, and the following four nodes can be any DAG structure, each having two incident edges. The last node serves as the output node, and each edge can take on one of eight operations. In this work, we perform our experiments on those three search spaces to evaluate our proposed SNAP flow.

The **search strategy** in NAS has been widely explored. There are well-known black-box optimizations, such as random selection with full training, reinforcement learning, evolution, Bayesian

Figure 2: SNAP, featuring operator description embedding and GCN-based performance predictor

optimization, Monte Carlo tree search, etc. With these search strategies, we still need to train the searched architecture and use the performance result to guide the search, which is a time-consuming process. To overcome the training time bottleneck, one-shot techniques were introduced as an **estimation strategy**. These techniques involve training a single (one-shot) supernet, which is an over-parameterized architecture that encompasses all possible architectures in the search space Cai et al. (2019); Liu et al. (2018c). Once the supernet is trained, each architecture in the search space can be evaluated by inheriting its weights from sampling the subnet within the supernet. Supernet design and training often become the performance bottleneck of these approaches. Supernet training typically dominates the NAS runtime.

## 2.2 ZERO-COST METHOD VS. MODEL-BASED PREDICTORS

To accelerate the search process, Zero-cost(ZC) proxies are proposed. They are lightweight metrics calculated on a network at initialization, often using just a single mini-batch of data. These methods evaluate intrinsic network properties without any parameter updates. For example, SNIP Lee et al. (2019b) assesses connection sensitivity; SynFlow Abdelfattah et al. (2021) computes a synaptic saliency score; TE-NAS Chen et al. (2021a) formulates neural networks as a Gaussian process and analyzes randomly-initialized architectures by the spectrum of the neural tangent kernel (NTK) Jacot et al. (2020); Xiao et al. (2020) and the number of linear regions in the input space.. While extremely fast, these proxies are often unstable and struggle to generalize across different search spaces.

Despite the inherent limitations of zero-cost proxies, the integration of model-based prediction has emerged as a pivotal component in guiding neural architecture search (NAS) algorithms. This approach is particularly useful when combined with Bayesian optimization and utilized as a subroutine Kandasamy et al. (2019); Shi et al. (2020). Various types of predictor models, including Gaussian processes (GP), multi-layer perceptron (MLP), long short-term memory (LSTM) networks, and graph neural networks (GNN), have been employed. Typically, as the algorithm progresses and a set of fully evaluated architectures becomes available, a meta-model is trained using architecture topology as features and validation accuracies as labels. This model is then used to predict the validation accuracy of yet-to-be-evaluated architectures. Notably, White et al. White et al. (2021) demonstrated that augmenting the model with Jacobian covariance as an additional feature can enhance its performance by up to 20%. Shen et al. Shen et al. (2023) further extended this approach by integrating zero-cost proxies into Bayesian optimization, resulting in 3-5× speedups over previous methods.

Existing model-based predictor approaches exhibit notable biases, limiting their effectiveness to specific search spaces and requiring fully evaluated architectures as labels. The high initialization time is also a concern. Dudziak et al. Łukasz Dudziak et al. (2021) attempted to address this issue by leveraging the model's binary relation and training a prediction model through iterative data selection. However, such models typically rely on fixed, one-hot encodings for operators, which prevents them from generalizing to new search spaces with previously unseen operations. GRAF Kadlecová et al. (2024) predictor demonstrated that using carefully engineered, interpretable graph features (like operation counts and path lengths) as input to a random forest model can also yield surprisingly strong performance predictions. While effective, this method relies on hand-crafting features for a given search space.

Table 1: Example of Operator Descriptive Sentences Vary in Length for Generating Embedding

| Operators | Short Sentences | Medium-length Sentences | Long Sentences |
|---|---|---|---|
| none | *"None"* | *"Doing nothing"* | *"A none operator that does nothing"* |
| skip_connect | *"Residual connection"* | *"Identity mapping to the next layer"* | *"A residual connection operator that adds identity mapping to the next layer"* |
| nor_conv_3x3 | *"Convolution 3x3"* | *"Convolution 3 by 3 kernel, Batchnorm, ReLU"* | *"A two-dimensional convolutional operator with a kernel size of 3 by 3 is applied, succeeded by a batch normalization layer, and followed by a rectified linear layer"* |
| nor_conv_1x1 | *"Convolution 1x1"* | *"Convolution 1 by 1 kernel, Batchnorm, ReLU"* | *"A two-dimensional convolutional operator with a kernel size of 1 by 1 is applied, succeeded by a batch normalization layer, and followed by a rectified linear layer"* |
| avg_pool_3x3 | *"Average pooling 3x3"* | *"Average pooling 3 by 3 kernel"* | *"A average pooling operator with a kernel size 3 by 3"* |

These limitations prompt us to explore a more robust operator encoding method that enables a pre-trained prediction model to operate effectively in any architecture search space and accommodate unseen operators. Essentially, we investigate the viability of relying solely on pre-trained model-based prediction as a universal zero-shot predictor for guiding searches across diverse architecture spaces with only one time training effort. Our SNAP directly learns the functional semantics of the operators from their textual descriptions, allowing it to generalize to entirely new operators and search spaces without manual feature engineering or retraining, offering a flexible and scalable path to a universal performance predictor.

## 3 SNAP: A Zero-shot Neural Predictor

In this work, we propose SNAP, a novel zero-shot neural predictor for neural architecture search (NAS). SNAP uniquely combines a transformer-based semantic embedding generator with a Graph Convolutional Network (GCN) predictor to accurately forecast architecture performance. Unlike traditional predictors, SNAP can effectively generalize to previously unseen operators and new search spaces without additional retraining or fine-tuning. Specifically, our transformer-based embedding generator captures meaningful semantic features from operator descriptions, while our GCN predictor—trained just once on an existing NAS benchmark—leverages these embeddings to predict the accuracy (or ranking) of candidate architectures. As illustrated in Figure 2, the predicted rankings, independent of new training data, guide the architecture search efficiently. Consequently, SNAP requires only a single initial training phase and seamlessly transfers its predictive capability to diverse new search spaces with no further training effort.

### 3.1 Architecture Representation

Various neural architecture search works represent their networks in the cell structure. For instance, DARTS Liu et al. (2018c), and NATS-Bench (NAS-Bench-201) Dong & Yang (2020) define a cell-based search space representing each architecture as a directed acyclic graph (DAG), with nodes representing the features. NAS-Bench-101 Ying et al. (2019) on the other hand utilizes nodes to represent layers (operators) and edges for forward dataflow propagation. BANANAS White et al. (2020) proposed a novel path-based encoding scheme and claimed it scales better than other methods. Additionally, Yan et al. Yan et al. (2021) propose a transformer-based encoding scheme with computation awareness. On contrary, our transformer-based neural architecture coding uses text-based DNN operator descriptions and sentence transformer.

In this work, we represent the DNN architecture candidates using DAG with nodes representing operators and edges corresponding to model data propagation flow. We then represent the graphs with adjacent matrix and operator node embeddings, which becomes data input for our graph convolution network (GCN)Kipf & Welling (2017) predictor. For example, consider a NAS-Bench-201 edge-labeled cell: edges (0→2: conv3x3, 1→2: skip). This converts to a node-labeled DAG: nodes [input, input, conv3x3, skip, output] with adjacency matrix connecting input nodes to operations, operations to output. To make all search space comply with this representation, SNAP unifies other cell-based search space representations as shown in Figure 2. NAS-Bench-101 was provided with the aforementioned architecture graphs, therefore no transformation is needed and was used as training data. Figure 2 shows an example on NAS-Bench-201. The final model architectures of each search space are obtained by stacking multiple repeated cells with some other predefined cells in between. The differences between different DNN architecture candidates are purely determined by the cell architecture (represented as graph), so we use the embedding of individual cell architectures (as graphs) to represent the entire DNN architecture. Operator node features are encoded using a transformer model with a fixed length embedding size. The details can be seen in the following section.

Table 2: Different Combination of Sentences Transformer Model and Embedding Sentence Length

| Model | Model Size (MB) | Embedding Size | Kendall's $\tau$ | | | Spearman's $\rho$ | | |
|---|---|---|---|---|---|---|---|---|
| | | | Short | Medium | Long | Short | Medium | Long |
| all-mpnet-base-v2 | 420 | 768 | 0.48 | 0.49 | 0.46 | 0.66 | 0.67 | 0.65 |
| MiniLM-L6-v2 | 80 | 384 | **0.56(0.60**[*]**)** | 0.44 | 0.40 | **0.76(0.80**[*]**)** | 0.62 | 0.56 |
| MiniLM-L6-v2-64 | 81 | 64 | 0.36 | 0.29 | 0.32 | 0.54 | 0.43 | 0.47 |

*Fine Tuned with Augmented DNN Operator Descriptions

## 3.2 OPERATOR EMBEDDING GENERATOR

In our approach, we encode model cell graphs using an adjacency matrix together with node embeddings. However, encoding operators demand special attention as it involves representing and distinguishing various deep learning operators. Previous methods, which typically rely on one-hot vectors for operator encoding, are deemed suboptimal and non-portable, especially when dealing with unseen search spaces and operators.

Recognizing that the names of operators inherently contain valuable information, we assert that the **operator name** alone can provide insight into the operation. For instance, the operator name "*CONV3x3-BN-ReLU*" suggests that it contains a two-dimensional convolution with a 3x3 kernel, followed by batch normalization and rectified linear activation. Therefore, we propose to construct a robust embedding model capable of extracting internal semantic information from operator names or their descriptive sentences in natural languages. For example, in the high-dimensional encoding space, operators like *conv3x3* are expected to be closer to *conv5x5* than to *maxpool3x3*. Additionally, if the embedding model comprehends one type of operator, it should readily extend its knowledge to similar operators with, for example, different kernel sizes.

Certain existing works have attempted to construct embedding vectors from words or sentences, such as GloVe Pennington et al. (2014), or employed character embeddings to capture fine-grained semantic and syntactic regularities. However, our earlier experiments indicated that these methods face challenges when dealing with previously unseen words, particularly operators in our case. Consequently, we have opted for Sentence Transformer Reimers & Gurevych (2019) as our primary method for generating desired operator embeddings. As illustrated in Figure 2, the Sentence Transformer utilizes siamese and triplet network structures to derive semantically meaningful sentence embeddings that can be compared using cosine similarity. A pooling operation is applied to the output of the pre-trained transformer model to obtain a fixed-size sentence embedding. We compute the mean of all output word vectors as the pooling strategy to generate the final operator embedding.

In our experiments, we explored different pre-trained sentence transformer models and varying lengths of descriptive sentences for operator embedding generation. Table 1 defines three categories of operator descriptions (short, medium, and long sentences) used as input for the embedding models. The embedding performance across these sentence variations on NAS-Bench-201 is shown in Table 2.

We examined three pre-trained sentence transformer models: *all-mpnet-base-v2*, *MiniLM-L6-v2*, and *MiniLM-L6-v2-64* (a PCA-downsampled version of MiniLM-L6-v2), each pretrained on the same dataset collection containing over 1 billion sentence pairs Reimers & Gurevych (2019). Training utilized a triplet objective loss function, designed to minimize the embedding distance between similar sentence pairs and maximize the distance between dissimilar pairs:

$$\mathcal{L} = \max(|s_a - s_p| - |s_a - s_n| + \epsilon, 0) \tag{1}$$

where $s_a$, $s_p$, and $s_n$ represent the embeddings of the anchor, positive, and negative sentences, respectively; $|\cdot|$ denotes Euclidean distance; and the margin $\epsilon$ is set to 1.

Table 2 shows that the *MiniLM-L6-v2* model with an embedding length of 384 achieves the highest correlation coefficients (Kendall's $\tau$ and Spearman's $\rho$) specifically when short operator sentences are used. We hypothesize this outcome arises because the semantic information required for embedding DNN operators is relatively straightforward and concise. Therefore, more elaborate or longer sentence descriptions may introduce redundant or unnecessary semantic detail, leading to embedding saturation or noise. Consequently, the combination of short sentence descriptions with the *MiniLM-L6-v2* embedding model was selected for subsequent fine-tuning and further experimentation.

To enhance the specificity of the embedding for neural architecture search, we fine-tuned the *MiniLM-L6-v2* model on a custom similarity dataset derived from PyTorch's `torch.nn` documentation. Operators were categorized by functionality (e.g., convolution, pooling, normalization, activation), and GPT-4o-generated augmented descriptions further diversified the linguistic formulations within each functional class. Training involved supervised similarity pairs labeled based on functional similarity, using a cosine similarity loss to guide the embedding alignment. Detailed methodology for this fine-tuning process is provided in the Appendix.

### 3.3 GCN PREDICTOR MODEL

After completing the universal architecture encoding and operator feature embedding, we employ a three-layer graph convolution network (GCN) Kipf & Welling (2017) as our prediction model. With the normalization trick, GCN can be defined as

$$H = X *_G G_\Theta = f(\bar{A}X\Theta) \tag{2}$$

where $\bar{A} = \tilde{D}^{-\frac{1}{2}}\tilde{A}\tilde{D}^{-\frac{1}{2}}$, $\tilde{A} = A + I_n$, and $\tilde{D}_{ii} = \sum_j \tilde{A}_{ij}$. To prevent overfitting to a particular training search space, we incorporate graph node normalization and weight decay techniques. Our overarching objective is to deliver a universally applicable pre-trained predictor model that requires no tuning for new search spaces. Consequently, the GCN predictor model is subject to heavy regularization. An additional crucial factor influencing our choice of GCN over other prediction models is its capability to handle vast differences in architectures. Given the varying dimensions of the adjacency matrix (from unseen search spaces) and operator matrix (from unseen operators), GCN emerges as a suitable choice, demonstrating flexibility in accommodating diverse architectural structures.

### 3.4 EVOLUTIONARY SEARCHING

Evolutionary and genetic algorithms have been commonly used to optimize the NAS White et al. (2023). To enhance the efficiency of the search, we adopt a similar approach in Algorithm 1. We first **initialize** the entire population of architecture candidates with continuous parameters in the first while loop, where we map all operators evenly into the value between 0 and 1. Then we discretized the random configurations into the desired NAS architectures. Then in the second while loop, **mutation** operation and **crossover** operation are performed to produce a new child. After generating all the offspring, the selection process will kick in using our proposed proxy, **GCN_Inference**, to select the elite candidates for the next generation. In the end, we select the final returned architecture as the search result.

---

**Algorithm 1** SNAP Search Algorithm

1: **Input:** $NP$ population size
2: $g \leftarrow 0$
3: **while** $|pop| < NP$ **do**
4:     $pop_i \leftarrow$ random_configuration()
5:     $pop_i' \leftarrow$ discretized_architecture($pop_i$)
6: **end while**
7: **while** $g < g_{max}$ **do**
8:     $V_g \leftarrow$ mutate($pop_g$)
9:     $U_g \leftarrow$ crossover($V_g, pop_g$)
10:    $U_g' \leftarrow$ discretized_population($U_g$)
11:    $fitness_g \leftarrow$ GCN_Inference($U_g'$)
12:    $pop_{g+1}, fitness_{g+1} \leftarrow$ select($pop_g, U_g$)
13: **end while**
14: **return** $Best\_Architecture$

---

## 4 EXPERIMENT AND RESULTS ANALYSIS

In our evaluation, we tested 12 NAS benchmarks similar to NAS-Bench-Suite-Zero Krishnakumar et al. (2022), including NAS-Bench-201 (CIFAR-10, CIFAR-100, and ImageNet16-120) Dong & Yang (2020), NAS-Bench-301 (CIFAR-10) Zela et al. (2022), and TransNAS-Bench-101 Micro (Jigsaw, Object Classification, Scene Classification, Autoencoder, Room Layout, Surface Normal, and Semantic Segmentation) Duan et al. (2021). We excluded NAS-Bench-101 Ying et al. (2019) from the comparison because it was used as training data for our GCN predictor. To prevent information leakage and ensure a fair comparison, we removed it from subsequent evaluations.

The GCN predictor consists of three graph convolutional layers with 64 hidden dimensions each, followed by batch normalization and ReLU activation. We use MSE regression loss, AdamW optimizer with learning rate 0.001, and StepLR scheduler (step_size=40, gamma=0.1). The sentence transformer operator embeddings are pre-computed and remain frozen during GCN training. The

model was trained for 150 epochs to ensure convergence. To minimize training cost and highlight the effectiveness of our novel architecture and operator encoding method in generalizing to unseen operators, the GCN was trained solely on NAS-Bench-101 (CIFAR-10) Ying et al. (2019).

## 4.1 GENERALIZABILITY ANALYSIS OF ZERO-SHOT METHODS ACROSS 12 BENCHMARKS

In Figure 3, we present the Spearman's $\rho$ correlation between each zero-shot method's predictions and the corresponding ground-truth performance across multiple benchmarks. While *Jacov* Mellor et al. (2021) and *Zen Score* Lin et al. (2021) show the highest correlations among existing proxies, SNAP consistently surpasses all methods.

Notably, SNAP retains robust predictive accuracy even in challenging benchmarks, such as the Autoencoder and Room Layout tasks from TransNAS-Bench-101-Micro Duan et al. (2021), where most zero-shot methods typically struggle. Furthermore, on the widely used

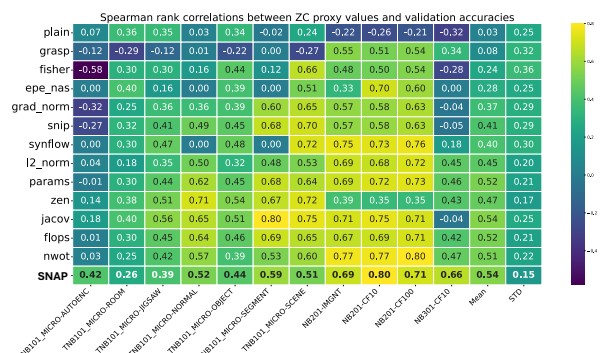

Figure 3: Spearman's $\rho$ rank between proxy values and ground truth accuracies, for 14 proxies and across 12 NAS benchmarks (the higher the better).

NAS-Bench-201 Dong & Yang (2020) benchmarks, SNAP outperforms existing proxies, including methods like *Snip* Lee et al. (2019a) and *Grasp* Wang et al. (2020). While these latter methods exhibit competitive results on NAS-Bench-201, they fail to generalize effectively to other benchmarks and can even be outperformed by simpler heuristics such as parameter counts (*Params*) or floating point operations (*FLOPs*).

Overall, these experimental outcomes underscore that SNAP achieves not only superior average predictive performance across diverse NAS benchmarks but also demonstrates consistently lower variance, highlighting its exceptional generalizability and portability.

## 4.2 INDEPENDENT ANALYSIS OF ZERO-SHOT METHODS

Combining multiple proxies can potentially enhance the accuracy and robustness of architecture performance predictions. However, not all proxies contribute unique information; some proxies may exhibit highly correlated rankings and therefore offer minimal complementary benefit. To explore this, we conducted a thorough assessment of proxy methods within the NAS-Bench-201 search space across three distinct datasets (CIFAR-10, CIFAR-100, and ImageNet16-120). Additionally, by examining the same search space across multiple tasks, we sought to identify whether certain proxies can provide universally appli-

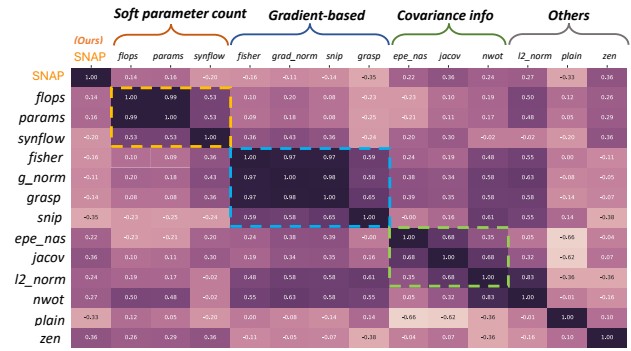

Figure 4: Spearman's $\rho$ correlation between proxy scores of CIFAR-10 on NAS-Bench-201.

cable, data- and task-independent rankings. Our analysis considered 14 proxies, including our proposed SNAP method. Specifically, we randomly sampled 1,000 architectures within NAS-Bench-201, computed each proxy's score per architecture, and then calculated Spearman's rank correlation among the proxies.

Table 3: Results of CIFAR-10, CIFAR-100 and ImageNet16-120 on NAS-Bench-201

| Name of Works | FLOPs (M) | Params (M) | Search Cost (GPU Hours) | CIFAR-10 Accuracy (%) | CIFAR-100 Accuracy (%) | ImageNet16-120 Accuracy (%) |
|---|---|---|---|---|---|---|
| μNASLiberis et al. (2021) | 7.78 | 0.073 | 552 | 86.49 | 58.30 | 27.80 |
| DARTSLiu et al. (2018c) | 82.49 | 0.587 | 3.02 | 88.32 | 67.78 | 34.60 |
| GDASDong & Yang (2019) | 117.88 | 0.83 | 8.03 | 93.36 | 69.64 | 38.87 |
| KNASXu et al. (2021) | 153.27 | 1.073 | 2.44 | 93.43 | 71.05 | 45.05 |
| NASWOTMellor et al. (2021) | 86.43 | 0.615 | 0.09 | 92.96 | 69.70 | 44.47 |
| TE-NASChen et al. (2021a) | 188.66 | 1.317 | 0.43 | 93.78 | 70.44 | 41.40 |
| **SNAP (ours)** | **113.95** | **0.802** | **0.01** | **93.75** | **70.64** | **44.97** |
| Ground Truth | 153.27 | 1.073 | - | 94.37 | 73.22 | 46.71 |

Our results reveal notable trends of high inter-correlations among certain proxies, particularly between *FLOPs* and parameter count (*Params*). This outcome aligns with intuitive expectations, as both proxies inherently measure aspects of computational complexity. Consequently, we extended our correlation analysis to cover all proxy pairs comprehensively, as illustrated by the correlation heatmap in Figure 8.

Consistent with previous literatureNing et al. (2021), *synflow* Abdelfattah et al. (2021) exhibited strong correlations with *FLOPs* and *Params*, likely due to its dependence on model complexity. Similarly, proxies derived from gradient saliency metrics, such as *grad_norm* Abdelfattah et al. (2021), *snip* Lee et al. (2019a), *grasp* Wang et al. (2020), *nwot* Mellor et al. (2021), and *fisher* Turner et al. (2020), also showed substantial mutual correlation. We observed similar high correlations between *epe_nas* Lopes et al. (2021) and the Jacobian covariance method (*jacov*) Mellor et al. (2021), likely due to their shared underlying analytical principles.

Interestingly, the *zen score* Lin et al. (2021) and our proposed SNAP predictor displayed the highest degree of independence relative to other evaluated proxies. This suggests that SNAP offers a fundamentally distinct perspective on neural architecture evaluation, potentially complementing and enriching existing proxy-based evaluations. Such independence indicates SNAP's potential to form powerful proxy combinations for more effective, generalized, and robust architecture performance predictions. We excluded some methods from our comparisons due to practical limitations: specifically, those requiring supernet training or lacking publicly available implementations Łukasz Dudziak et al. (2021); Cai et al. (2019); Shi et al. (2020); Xu et al. (2020).

### 4.3 NAS Result on NAS-Bench-201

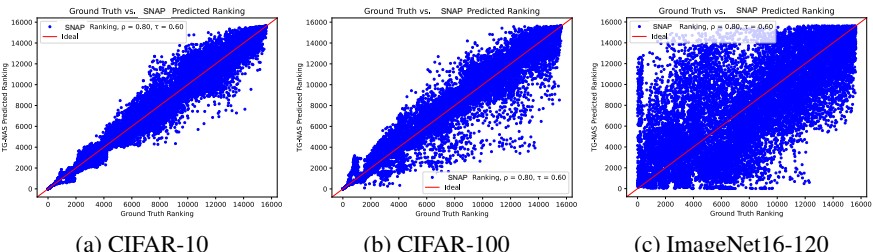

(a) CIFAR-10      (b) CIFAR-100      (c) ImageNet16-120

Figure 5: SNAP vs. Ground Truth ranking on NAS-Bench-201 Space

Figure 7 demonstrates that our SNAP is highly positively correlated with the architecture's accuracy ranking. Additionally, as illustrated in Table 3, our SNAP outperforms the majority of zero-shot NAS approaches and is comparable to the state-of-the-art TE-NAS results, achieving 93.75% top-1 accuracy on CIFAR-10, while consuming only a fraction of the searching time of prior works.

Notably, despite efficiency claims of zero-shot NAS methods over conventional NAS due to avoiding the training of sampled architectures, substantial variations in computational costs and search times persist among these methods. For instance, TE-NAS Chen et al. (2021a) requires over 4 GPU-hours, while ZiCo Li et al. (2023) demands more than 10 GPU-hours for an ImageNet search on the DARTS space. These longer search times primarily arise because several zero-shot methods necessitate multiple forward or forward-backward passes for result stabilization, often making them data-dependent and thus limiting their generalizability.

Table 4: Comparison with Recent NAS Works on ImageNet with the Mobile Setting

| Name of Works | Test Accuracy (%) Top-1 | Top-5 | Search Cost (GPU Days) | Params (M) | Search Method |
|---|---|---|---|---|---|
| PNAS Liu et al. (2018a) | 74.2 | 91.9 | 225 | 5.1 | Bayesian Optimization |
| AmoebaNet-C Real et al. (2019) | 75.7 | 92.4 | 3150 | 6.4 | Evolution |
| NASNet-A Zoph et al. (2018) | 74.0 | 91.6 | 2000+ | 5.3 | Reinforcement Learning |
| DARTS Liu et al. (2018c) | 73.3 | 91.3 | 4.0 | 4.7 | Gradient-based |
| SNAS Xie et al. (2018) | 72.7 | 91.8 | 1.5 | 4.3 | Gradient-based |
| BayesNAS Zhou et al. (2019) | 73.5 | 91.1 | 0.2 | 3.9 | Gradient-based |
| ProxylessNAS Cai et al. (2019) | 75.1 | 92.5 | 8.3 | 7.1 | Gradient-based |
| TE-NAS Dong & Yang (2019) | 73.8 | 91.7 | 0.05 | 6.3 | Theoretical Analysis |
| PC-DARTS Xu et al. (2020) | 74.9 | 92.2 | 0.1 | 5.3 | Gradient-based |
| PNASNet-5 Liu et al. (2018a) | 74.2 | 91.9 | 45 | 5.1 | Model-based Predictor |
| GHN Zhang et al. (2018) | 73.0 | 91.3 | 0.84 | 5.7 | Model-based Predictor |
| NAONet Luo et al. (2018) | 74.3 | 91.8 | 200 | 11.35 | Model-based Predictor |
| GeNAS Jeong et al. (2023) | 75.3 | 92.4 | 0.4 | 5.3 | Model-based Predictor |
| SemiNAS Luo et al. (2020b) | 76.5 | 93.2 | 4 | 6.3 | Model-based Predictor |
| CTNAS Chen et al. (2021b) | 77.3 | 93.4 | 50.1 | - | Model-based Predictor |
| **SNAP (ours)** | **74.9** | **92.2** | **0.0014** | **5.6** | **Model-based Predictor** |

In contrast, our proposed SNAP method completes the search in approximately 40 seconds, achieving more than a $10\times$ speedup relative to other zero-shot methods, owing to its lightweight, data-independent predictive framework. Although our predictor requires an initial computational investment for training on NASBench-101 (approximately 1.5 GPU-hours), this is a one-time cost that enables subsequent performance predictions without additional retraining or tuning across new search spaces and tasks.

### 4.4 NAS Result for ImageNet on the DARTS Space

For the DARTS search space, the final discovered cell architecture is provided in the Appendix. After determining the optimal cell, we constructed the final network by stacking 14 cells, initializing with a channel count of 48. Performance comparisons of SNAP-discovered architectures against recent NAS methods are summarized in Table 7. SNAP achieves competitive top-1 and top-5 accuracies of 74.9% and 92.2%, respectively. Remarkably, SNAP delivers a substantial efficiency advantage, offering more than $35\times$ faster search times compared to previous state-of-the-art zero-shot methods and model-based predictors, completing the entire search process in less than two minutes on a single NVIDIA RTX 4090 GPU. SNAP's ability to generalize predictions across diverse, previously unseen search spaces without additional fine-tuning represents a significant advancement over existing methods.

### 4.5 Limitations

Our SNAP predictor requires an initial training phase on NASBench-101, which involves a one-time computational cost. Although it generalizes well to previously unseen operators, it has so far been limited to cell-based search spaces. Extending its applicability to fundamentally different architecture types remains an open area for future research.

## 5 Conclusion

In this work, we propose SNAP, a zero-shot neural predictor for architecture search that is broadly applicable to new search spaces which containing previously unseen operators. SNAP integrates text-based operator descriptions—processed by a fine-tuned sentence transformer—with a graph convolutional network (GCN) predictor to enable architecture performance estimation. It offers key advantages in robustness, generalizability, proxy independence, and cost-effectiveness. Our experiments demonstrate that SNAP consistently outperforms existing proxies across a wide range of NAS benchmarks, establishing it as a strong foundational component for efficient architecture search. SNAP achieves more than $35\times$ improvement in search efficiency over prior state-of-the-art methods. Notably, it discover competitive models with 93.75% CIFAR-10 accuracy on the NAS-Bench-201 space and 74.9% ImageNet top-1 accuracy on the DARTS space.

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

## A    APPENDIX

This documentation includes technical appendix such as additional figures or tables, and more detailed analyses of experiments presented in the paper *SNAP: Generalizable Zero-Shot Prediction of Neural Architecture Performance via Semantic Embedding and Graph Learning*

## B    GCN PREDICTOR FUNCTIONAL VALIDATION

To evaluate the applicability of the constructed GCN predictor, we partition the NAS-Bench-101 space into train/validation splits ranging from 90% to 1%. As shown in Figure 6, the predictor maintains strong performance even when trained on as little as 1% of the architecture data, as evidenced by high Kendall's $\tau$ and Spearman's $\rho$ correlation coefficients. Table 5 further compares these correlation scores with those of existing zero shot methods on NAS-Bench-201, highlighting the superior performance of our approach. Additionally, we investigate the impact of varying the number of GCN layers and training hyperparameters, with results summarized in Table 6 as an ablation study.

Table 5: *Kendall's* $\tau$ and *Spearman's* $\rho$ correlation between various zero-shot methods on NAS-Bench-201

| Metric | Params Pham et al. (2018) | FLOPs Pham et al. (2018) | SNIP Lee et al. (2019b) | Fisher Turner et al. (2020) | Synflow Tan et al. (2019) | Zen-score Lin et al. (2021) | Grad-norm Abdelfattah et al. (2021) | SNAP (ours) |
|---|---|---|---|---|---|---|---|---|
| *Kendall's* $\tau$ | 0.55 | 0.54 | 0.41 | 0.22 | 0.54 | 0.29 | 0.37 | **0.60** |
| *Spearman's* $\rho$ | 0.74 | 0.73 | 0.58 | 0.36 | 0.73 | 0.38 | 0.54 | **0.80** |

## C    ADDITIONAL RANKING FIGURES OF SNAP VS. GROUND TRUTH ON NAS-BENCH-201

In this section, we present additional accuracy and ranking vs. ground truth figures for CIFAR-10, CIFAR-100, and ImageNet16-120 datasets on the NAS-bench-201 space. The SNAP GCN model utilized here is trained using the entire NAS-bench-101 benchmark with CIFAR-10 accuracy results only. Figure 7 demonstrates the correlation between our SNAP predictions and the raw model accuracies, revealing a high positive correlation. This result is consistent with our previous ranking experiments. We believe that incorporating more diverse benchmarks, including additional evaluation results from different datasets on the same architectures, will enhance the stability and generalizability of our SNAP approach.

Table 6: Effect of Model Settings on Kendall's $\tau$ Ranking

| GCN Layers | Weight Decay | Sentence Length | Embedding Size | Kendall's $\tau$ |
|---|---|---|---|---|
| 4 | $1e^{-4}$ | Long | 384 | 0.487 |
| 4 | $1e^{-5}$ | Long | 384 | 0.401 |
| 4 | $1e^{-6}$ | Long | 384 | 0.496 |
| 4 | $1e^{-4}$ | Short | 384 | 0.495 |
| 4 | $1e^{-5}$ | Short | 384 | 0.454 |
| 4 | $1e^{-6}$ | Short | 384 | 0.497 |
| 4 | $1e^{-4}$ | Long | 768 | 0.433 |
| 4 | $1e^{-5}$ | Long | 768 | 0.421 |
| 4 | $1e^{-6}$ | Long | 768 | 0.454 |
| 4 | $1e^{-4}$ | Short | 768 | 0.433 |
| 4 | $1e^{-5}$ | Short | 768 | 0.421 |
| 4 | $1e^{-6}$ | Short | 768 | 0.454 |
| 3 | $1e^{-4}$ | Long | 384 | 0.483 |
| 3 | $1e^{-5}$ | Long | 384 | 0.566 |
| 3 | $1e^{-6}$ | Long | 384 | 0.577 |
| 3 | $1e^{-4}$ | Short | 384 | 0.601 |
| 3 | $1e^{-5}$ | Short | 384 | 0.599 |
| 3 | $1e^{-6}$ | Short | 384 | 0.598 |
| 3 | $1e^{-4}$ | Long | 384 | 0.516 |
| 3 | $1e^{-5}$ | Long | 384 | 0.557 |
| 3 | $1e^{-6}$ | Long | 384 | 0.594 |
| 3 | $1e^{-4}$ | Short | 768 | 0.538 |
| 3 | $1e^{-5}$ | Short | 768 | 0.545 |
| 3 | $1e^{-6}$ | Short | 768 | 0.513 |

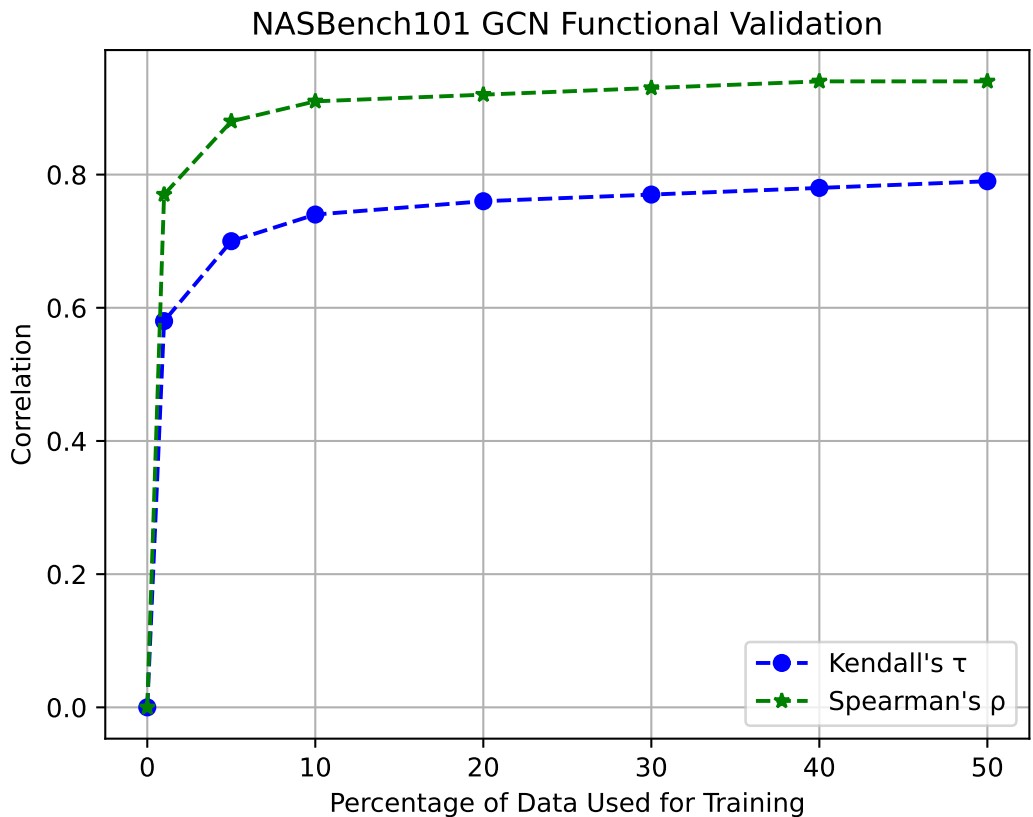

Figure 6: GCN Predictor Functional Validation on NAS-Bench-101 Benchmark

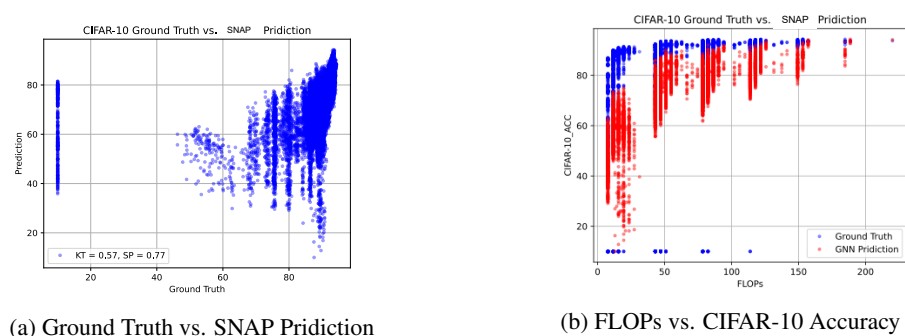

(a) Ground Truth vs. SNAP Pridiction

(b) FLOPs vs. CIFAR-10 Accuracy

Figure 7: SNAP Accuracy Correlation Evaluation with Ground Truth on NAS-Bench-201 Space

## D    ADDITIONAL CORRELATION FIGURES WITH OTHER ZERO-SHOT METHODS OF CIFAR-100 AND IMAGENET16-120 DATASET ON NAS-BENCH-201

In this section, we present heatmaps showing the correlations between pairs of popular proxies, calculated using the *CIFAR-100* and *ImageNet16-120* datasets as shown in figure 8. These heatmaps reveal correlation trends that are consistent with those observed in the *CIFAR-10* results.

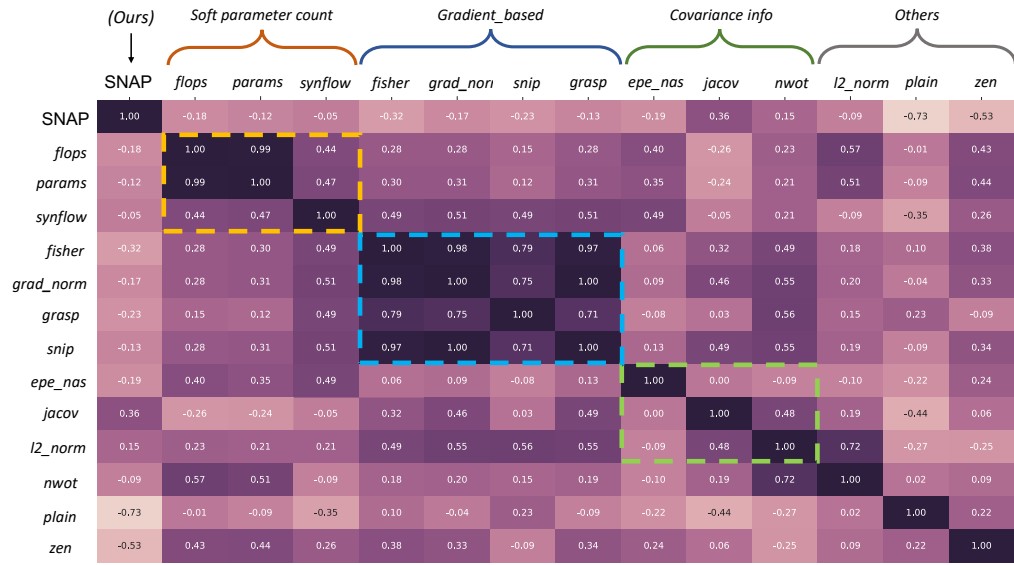

(a) Spearman's $\rho$ Correlation of CIFAR-100

(b) Spearman's $\rho$ Correlation of ImageNet16-120

Figure 8: Spearman's $\rho$ Correlation for all Pairs of Zero-shot Methods of CIFAR-100 and ImageNet16-120 on NAS-Bench-201

# E  ADDITIONAL COMPARISON WITH MORE PREDICTOR-BASED NAS WORKS

In this section, we expand our comparison to include other NAS works involving training-based predictors. However, it's worth noting that the formulation and training of these predictor models differ from ours. They often require iterative training with new golden truth samples during the search process. Consequently, they are not zero-shot, highly bound to specific search spaces, lack general applicability, and remain costly to employ. To address this limitation, we propose the Transformer-based operator embedding method. This approach enables our predictor model to be decoupled from search spaces, thereby allowing our method to serve as a general zero-shot predictor model. As illustrated in Table 7, our SNAP achieves significantly higher search efficiency compared to other

predictor-based NAS Works. Specifically, our method demonstrates search efficiency ranging from 71 times to $1.4 \times 10^5$ times better, highlighting its effectiveness and scalability in NAS tasks.

Table 7: Comparison with additional predictor-based NAS works on ImageNet. The use of "†" indicates that the search is not conducted on the standard DARTS space, and we compare their results with similar mobile settings.

| Name of Works | Test Accuracy (%) | | Search Cost (GPU Days) | Params (M) | Search Method |
|---|---|---|---|---|---|
| | Top-1 | Top-5 | | | |
| NGELi et al. (2020) | 74.7 | 92.1 | 0.1 | 5.1 | Model-based Predictor |
| GHNZhang et al. (2018) | 73.0 | 91.3 | 0.84 | 5.7 | Model-based Predictor |
| NAONetLuo et al. (2018) | 74.3 | 91.8 | 200 | 11.35 | Model-based Predictor |
| PNASNet-5Liu et al. (2018a) | 74.2 | 91.9 | 45 | 5.1 | Model-based Predictor |
| GBDT-NAS-3S† Luo et al. (2020a) | 76.5 | 93.2 | 4 | 6.4 | Model-based Predictor |
| CTNAS† Chen et al. (2021b) | 77.3 | 93.4 | 50.1 | - | Model-based Predictor |
| SemiNAS† Luo et al. (2020b) | 76.5 | 93.2 | 4 | 6.3 | Model-based Predictor |
| ZenNet-400M† Lin et al. (2021) | 78.0 | - | 0.5 | 5.7 | Zero Shot |
| ZiCo-450M† Li et al. (2023) | 78.1 | - | 0.4 | 4.5 | Zero Shot |
| **SNAP (ours)** | **74.5** | **91.9** | **0.0014** | **5.6** | **Model-based Predictor** |

## F  SENTENCE TRANSFORMER FINETUNE SETUP

To formulate supervised training pairs, we define three types of similarity relations:

- Positive pairs (similarity = 1.0): Descriptions from the same class, including original and GPT-augmented versions (e.g., torch.nn.Conv2d, Pytorch offical description, and GPT4o augmented descriptions).

- Related pairs (similarity = 0.7): Descriptions from different classes within the same category (e.g., torch.nn.Conv2d and torch.nn.ConvTranspose2d).

- Unrelated pairs (similarity = 0.0): Descriptions from distinct functional categories (e.g., torch.nn.Conv2d vs. torch.nn.BatchNorm2d).

We fine-tune the model using cosine similarity loss, which encourages the embedding space to reflect semantic relationships: similar operators are embedded closer together, while unrelated ones are pushed apart. As shown in Table 8, this fine-tuning significantly improves the model's ability to differentiate operator semantics. For example, the similarity score between conv2x2 and "A 2D conv layer with a 2x2 kernel" increases from 0.6052 to 0.8847, while the unrelated pair conv2x2 and maxpool drops from 0.2102 to 0.0275. This indicates the model learns to align functionally related operators while effectively distinguishing dissimilar ones.

Table 8: Operator Description's Similarity Comparison Before and After Finetuning

| Operators | Compared Operators/Description | Similarity before Finetune | Similarity after Finetune |
|---|---|---|---|
| conv2x2 | maxpool | 0.2102 | 0.0275 |
| conv2x2 | A 2D conv layer with a 2x2 kernel. | 0.6052 | 0.8847 |
| nn_Dropout | nn_BatchNorm2d | 0.4629 | -0.1160 |
| skip connection | residual | 0.0584 | 0.9113 |
| maxpool | avgpool | 0.2907 | 0.6741 |
| maxpool | conv2x2 | 0.2277 | 0.0011 |

## G  FINAL SEARCHED CELL ARCHITECTURE

Our final search cell architecture on DARTS space can be found in figure 9

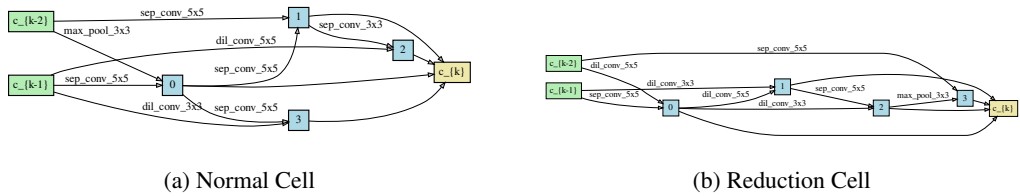

(a) Normal Cell

(b) Reduction Cell

Figure 9: Discovered Architecture on the DARTS Space

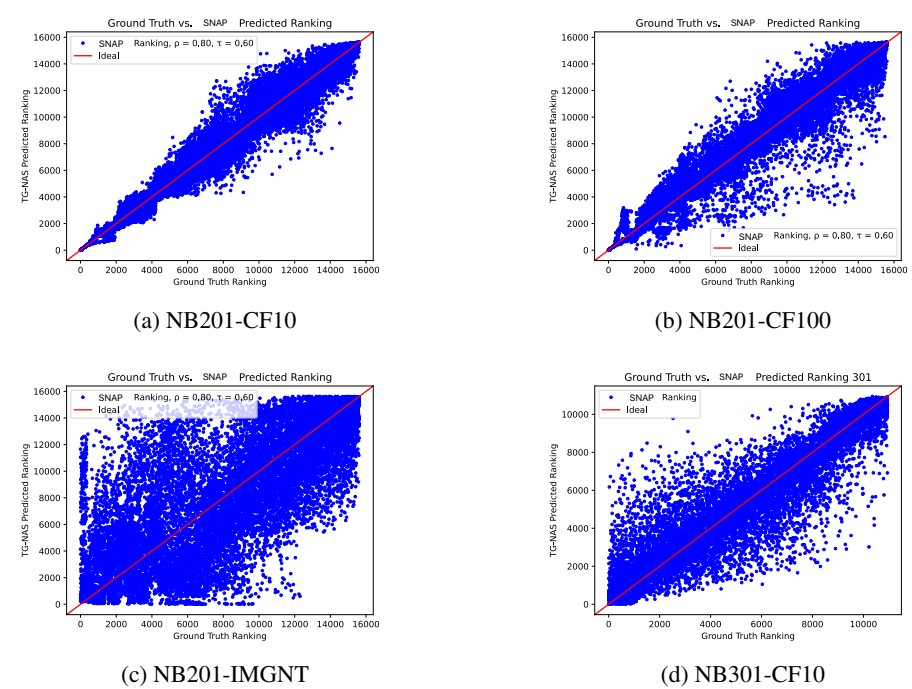

(a) NB201-CF10

(b) NB201-CF100

(c) NB201-IMGNT

(d) NB301-CF10

Figure 10: Ground Truth vs Predicted Results (Part 1 of 2)

## H    GROUND TRUTH AND PREDICTED RESULT COMPARISON

In Figure 10 and Figure 11 we compare SNAP's predicted ranking against the true ranking across ten benchmark tasks. The result on NAS-Bench-201 Cifar-10 and Cifar-100 exhibit very tight clustering, indicating good ranking fidelity. While the result for NAS-Bench-201 Imagenet shows relatively high variance, its Spearman's $\rho$ correlation is high as we have shown in the main paper. Overall, SNAP consistently captures the relative ordering of architectures across both large-scale image-classification benchmarks and fine-grained micro-benchmarks.

## I    CODE

Our code is provided in a dedicated .zip file.

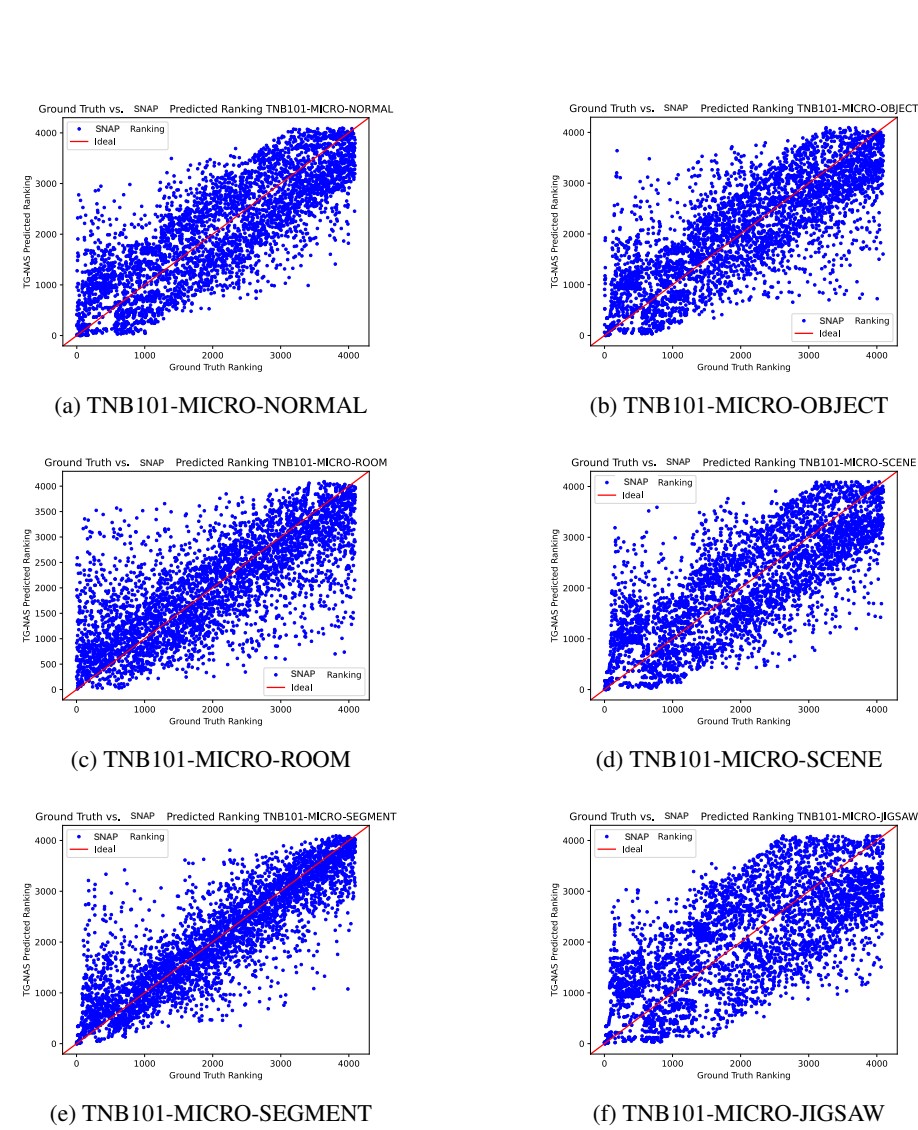

(a) TNB101-MICRO-NORMAL

(b) TNB101-MICRO-OBJECT

(c) TNB101-MICRO-ROOM

(d) TNB101-MICRO-SCENE

(e) TNB101-MICRO-SEGMENT

(f) TNB101-MICRO-JIGSAW

Figure 11: Ground Truth vs Predicted Results (Part 2 of 2)

