# OpenReview forum: "SNAP: Generalizable Zero-Shot Prediction of Neural Architecture Performance via Semantic Embedding and Graph Learning"
_ICLR.cc/2026/Conference — ICLR 2026 Conference Withdrawn Submission_

### Official Review · Reviewer_YatH · 2025-10-23

**Soundness:** 2
**Presentation:** 2
**Contribution:** 2
**Rating:** 2
**Confidence:** 5

**Summary:**

This paper introduces SNAP, a zero-shot neural architecture performance predictor that generalizes across unseen operators and search spaces without retraining. It combines a transformer-based semantic embedding of operator descriptions with a GCN to predict architecture performance. Trained only once on NAS-Bench-101, SNAP achieves state-of-the-art rank correlation on 12 NAS benchmarks and significantly improves search efficiency.

**Strengths:**

+ The paper is well written and easy to understand. The visualizations are nicely done.

+ The proposed method leverages fine-tuned sentence transformers to encode operator descriptions, enabling the model to understand and generalize to new or unseen operators based on their functional semantics.

**Weaknesses:**

+ The architecture of the proposed performance predictor lacks of novelty. To be more specific, the architecture is simply constructed by sequentially placing transformer and GCN, which are widely used in the community of performance predictor. As a result, the technical contribution of this work is limited.

+ Comparing with the zero-cost proxies, the proposed performance predictor still needs to be trained on the existing benchmark, i.e., NAS-Bench-101. Therefore, directly comparing the search cost with zero-cost proxies is unfair. The search cost of SNAP needs to include the training time of the performance predictor and the training time for the architectures used in NAS-Bench-101. In this way, the search cost of SNAP no longer has the advantage.

+ The motivation lacks proper evidences to support. Some empirical results are beneficial to support the claim “existing zero-cost proxies underperform simple heuristics, suffer from data dependency, and generalize poorly across diverse search spaces”.

+ The format of references needs to be adjusted. Please use \citep{} instead of \cite{}.

**Questions:**

Please refer to the Weaknesses part.

---

### Official Review · Reviewer_tKzr · 2025-10-24

**Soundness:** 2
**Presentation:** 2
**Contribution:** 1
**Rating:** 2
**Confidence:** 5

**Summary:**

This paper presents SNAP, a zero-shot performance predictor which leverages transformer-based semantic embeddings of operator descriptions and a graph convolutional network. After a single training phase on NAS-Bench-101, SNAP generalizes effectively to new search spaces and unseen operators without further fine-tuning. It demonstrates superior rank correlation across multiple benchmarks and enables highly efficient architecture discovery, achieving competitive accuracy on CIFAR-10 and ImageNet with significantly reduced search time.

**Strengths:**

+ The framework of the proposed method is clearly presented and easy to understand.

+ The paper has rich visualization results in terms of the correlation.

**Weaknesses:**

+ The performance predictor is trained only on NAS-Bench-101, raising concerns about its generalizability to other search spaces without further validation across multiple training sources. Why not train the performance predictor on NAS-Bench-201 or NAS-Bench-NLP.

+ The use of a transformer and GCN for performance prediction is common in the field, offering little innovation in model design compared to existing predictors.

+ The reported search efficiency ignores the substantial computational cost of training architectures in NAS-Bench-101, misleadingly presenting SNAP as more efficient than it is.

+ Despite claims of zero-shot generalization, the predictor may still be biased toward the structural and operational distribution of NAS-Bench-101, affecting true cross-space performance.

**Questions:**

+ How does SNAP's use of semantic embeddings from operator descriptions enable generalization to unseen search spaces without retraining?

+ Why do existing zero-cost proxies underperform simple heuristics, suffer from data dependency, and generalize poorly across diverse search spaces? I’d like to see more evidence which can support this claim.

---

### Official Review · Reviewer_Wxe5 · 2025-10-24

**Soundness:** 3
**Presentation:** 1
**Contribution:** 1
**Rating:** 2
**Confidence:** 5

**Summary:**

This paper proposes SNAP, short for Semantic Neural Architecture Predictor.
SNAP is a cross-domain predictor for Neural Architecture Search (NAS) that consists of using a language model to form semantic embeddings of architectures from different search spaces from graph (directed acyclic graph; DAG) encodings. SNAP is evaluated across several NAS Benchmarks and compared to other similar predictors like Zero-Cost Proxies.

**Strengths:**

The evaluation is not limited to just old NAS benchmarks like 101, 201 and 301. Also, interproxy correlation is calculated.

**Weaknesses:**

The formatting of this manuscript falls below the standard expected at ICLR. In particular, the citation style is inconsistent and often incorrect. The manuscript frequently uses constructions like “Author et. al, year” rather than the proper format “(Author et al., year),” which suggests either unfamiliarity with LaTeX citation commands such as \cite, \citet, and \citep, or a lack of attention to detail in preparing the submission. This detracts from the readability and professionalism of the paper.

Second, the figures vary significantly in quality. While Figures 1 and 2 are reasonably well-presented, Figures 3, 4 (especially), and 5 suffer from extremely small fonts and inconsistent formatting, making them difficult to interpret without zooming in. Improving figure clarity and consistency would greatly enhance the manuscript’s accessibility.

Third, the related work section of the paper is extremely sparse. For instance, in line 107 the authors list several types of search strategies for NAS but no citations, e.g., gradient based [1, 2], Random Search [3], MCTS [4], RL [5], Bayesian [6], etc. Several of these methods would make for good comparisons in Tabs. 3 and 4 especially, as they are more recent than the numerous pre-2020 papers the authors cite.

The above issues, however, can be addressed by extensive, compulsory revisions and additional experiments, which ICLR allows.

The deeper issue is that the paper's novelty is narrow and limited. The motivation regarding prediction across search spaces has been solved for cell-based NAS [7] and there are other works that extend such ideas beyond cell-based NAS without requiring a cumbersome semantic textual encoding of the architecture. This means the authors have a difficult challenge of overcoming why their method, which involves the textual language domain, would be superior to an existing cross-domain predictor. Unfortunately, they have not shown they are even aware of these contributions, yet propose something more burdensomely complex.

Additionally, the experimental results are not that impressive. Despite the methods heavy complexity, as Fig. 3 shows it is only 0.02 or 0.03 spearman rho better than FLOPs or nwot on average, respectively, which is not justifiable. Also the caption of Fig. 3 is incorrect because fisher is really the best proxy on TNB101_Micro-Autoenc you just multiply the prediction by -1 and get a spearman rho of 0.58 which is head and shoulders better than anything else. The reviewer also notes the author's mistake of bolding their method several times in Fig. 3 when it is not the best, e.g., TNB101_Micro-Segment or NB201-IMGNT. These concerns are further reinforced by Fig. 5, which is more to the paper's detriment.

Finally, we look to the downstream NAS results. As Tabs. 3 and 4 show, the authors only consider safe, but stale NAS benchmarks which  is not really sufficient and even then does not bring clear wins in terms of performance, i.e., in Tab. 3 SNAP loses to TE-NAS on CIFAR-10, loses to KNAS on CIFAR-100 and loses to NASWOT on ImageNet16-120. To steel man the author's case, while SNAP has the lowest search cost of 0.01 GPU hours, this is only really an advantage next to KNAS which uses 2.44 hours (yet gets significantly higher performance which justifies the extra GPU cost in the long run), while the GPU cost gain compared to TE-NAS and NASWOT, both <1 GPU hr, is not significant.

**Questions:**

N/A

References:

[1] DARTS: DIFFERENTIABLE ARCHITECTURE SEARCH - ICLR 2019

[2] Geometry-Aware Gradient Algorithms for Neural Architecture Search - ICLR 2021

[3] Random Search and Reproducibility for Neural Architecture Search - UAI 2019

[4] AlphaX: eXploring Neural Architectures with Deep Neural Networks and Monte Carlo Tree Search - AAAI-19

[5] L2NAS: Learning to Optimize Neural Architectures via Continuous-Action Reinforcement Learning - CIKM'21

[6] Interpretable Neural Architecture Search via Bayesian Optimisation with Weisfeiler-Lehman Kernels - ICLR 2021

[7] Bridge the Gap Between Architecture Spaces via A Cross-Domain Predictor - NeurIPS'22

---

### Official Review · Reviewer_kwWh · 2025-10-31

**Soundness:** 3
**Presentation:** 3
**Contribution:** 2
**Rating:** 6
**Confidence:** 2

**Summary:**

This paper introduces **SNAP**, a zero-shot neural architecture performance predictor that combines **semantic operator embeddings** generated via sentence-transformers with a **GCN-based architecture encoder**. The predictor is trained once on NAS-Bench-101 and then generalizes to unseen operators and search spaces without fine-tuning. Experiments across **12 NAS benchmarks** show that SNAP consistently outperforms zero-cost proxies, achieves competitive results to state-of-the-art NAS methods, and completes search within seconds.

The idea of injecting natural-language semantics into NAS representation is intuitive and addresses limitations of one-hot operator encodings for generalizable NAS.

**Strengths:**

1. **Addresses a meaningful challenge:** generalizable zero-shot NAS without re-training or data.
2. **Novel operator representation:** uses sentence embeddings to encode operator semantics rather than one-hot encoding.
3. **Strong empirical validation:** evaluated on NAS-Bench-201, NAS-Bench-301, and TransNAS-Bench-101 with consistent superiority.
4. **Practical and efficient:** search completes in ~40 seconds, significantly faster than TE-NAS, ZiCo, and other recent methods.
5. **Clear model architecture & ablations:** particularly the operator embedding analysis and GCN architecture study.

**Weaknesses:**

1. **Conceptual novelty is moderate:** core idea builds on predictor-based NAS + GNN, adding NLP embeddings; not fundamentally rethinking NAS formulation.
2. **Limited architectural scope:** currently validated only on cell-based CNN search space, limiting claims of universality.
3. **Weak analysis of semantic embedding quality:** lacks visualization or metrics demonstrating semantic hierarchy or functional clustering.
4. **Relies on NAS-Bench-101 exclusively for training:** generalization to ViT or hybrid spaces remains untested.
5. **Positioning with respect to recent LLM-NAS literature is insufficient.**

**Questions:**

1. How does SNAP handle more complex operators, such as attention, dynamic kernels, or mixed operations?
2. Does the model remain robust if operator descriptions are noisy or partially incorrect?
3. Could code-based embeddings or docstring parsing outperform natural-language operator names?
4. Can this approach be extended to architectures without repeated cells (e.g., ViT, OFA)?

---

### Note · Authors · 2025-11-23

I have read and agree with the venue's withdrawal policy on behalf of myself and my co-authors.